# Pre-Protection and Mechanism of Crude Extracts from *Dioscorea alata* L. on H_2_O_2_-Induced IPEC-J2 Cells Oxidative Damage

**DOI:** 10.3390/ani13081401

**Published:** 2023-04-19

**Authors:** Yanhong Yun, Huiyu Shi, Yanyu Wang, Fengyuan Yang, Yuanxin Zhang, Haibo Feng, Junpu Chen, Xuemei Wang

**Affiliations:** College of Animal Science and Technology, Hainan University, Haikou 570228, China

**Keywords:** *Dioscorea alata* L., proanthocyanidins, anthocyanins, IPEC-J2 cells, H_2_O_2_, pre-protection

## Abstract

**Simple Summary:**

The *Dioscorea alata* L. is known to contain abundant proanthocyanidins and anthocyanins, which exhibit potent antioxidant properties capable of scavenging free radicals. Additionally, these compounds have been shown to effectively promote the proliferation of IPEC-J2 cells within a pharmacotoxicity range. Conversely, exposure to H_2_O_2_ can damage IPEC-J2 cells and trigger increased activity of cellular antioxidant enzymes. This study investigated the impact of crude extract from *Dioscorea alata* L. on the antioxidant capacity of IPEC-J2 cells and explored the pre-protective mechanism of *Dioscorea alata* L. against oxidative damage induced by H_2_O_2_ through an examination of its effect on the NF-κB inflammatory signaling pathway.

**Abstract:**

The purple tubers of *Dioscorea alata* L. have been found to contain a variety of bioactive chemical components, including anthocyanins, which make it significant to investigate the pre-protective effects of *Dioscorea alata* L. and its crude extracts on cells prior to oxidative stress. To establish a suitable oxidative damage model, an injured model of IPEC-J2 cells was created using H_2_O_2_ as the oxidant. Specifically, when the concentration of H_2_O_2_ was 120 μmol/L and the injured time was 8 h, the survival rate of cells decreased to approximately 70%, and the cells exhibited a noticeable oxidative stress reaction. Moreover, the crude extracts of *Dioscorea alata* L. demonstrated beneficial pre-protective effects on IPEC-J2 cells by increasing the total antioxidant capacity (T-AOC) and catalase (CAT) activities, augmenting the expression of total superoxide dismutase (T-SOD) and its genes, reducing the content of malondialdehyde (MDA) and the activity of glutathione peroxidase (GSH-P_X_) and its expression of genes, and promoting the expression of glucose transporter SGLT1 gene while reducing that of GULT2 gene, thereby facilitating the entry of anthocyanins into cells. In addition, the 50 μg/mL crude extracts effectively inhibited the phosphorylation of IκB and the p65 protein, thus reducing cellular oxidative stress. Given these findings, *Dioscorea alata* L. can be considered a natural antioxidant for practical breeding and production purposes, with an optimal concentration of crude extracts in this experiment being 50 μg/mL.

## 1. Introduction

Gastrointestinal diseases represent a common health issue in the swine population worldwide. In a study conducted by Furbeyre et al. [1], it was conclusively demonstrated that after weaning, piglets experience a severe and potentially debilitating intestinal oxidative stress response, which in turn leads to a significant increase in the incidence of diarrhea, as well as a reduction in feed intake and production performance. Pathogenic factors such as lipopolysaccharides (LPS) and hydrogen peroxide (H_2_O_2_), produced by pathogenic bacteria such as Salmonella and *Escherichia coli*, play a key role in the onset of gastrointestinal diseases. In particular, *Escherichia coli* induces various gastrointestinal diseases in swines, with Enterotoxin Pathogenic Escherichia coli being the most common type induced by environmental factors. This type of bacteria triggers the production of a large number of oxidative inducers, such as reactive oxygen species (ROS), and a significant amount of H_2_O_2_ in the intestinal tracts of swines, making them more susceptible to diseases. Newborn and weaned piglets are particularly susceptible to these diseases, as Luppi et al. [2] have shown. H_2_O_2_ is one of the most commonly used oxidants, and in a study conducted by Weber et al. [3], it was observed that when a suitable amount of H_2_O_2_ was administered to rat pancreatic acinar AR42J cells, it activated the cellular antioxidant system, as well as the defense systems of nuclear transcription factor κB (NF-κB), and inhibited cellular cycle progression.

In addition to its starch and nutritional content, *Dioscorea alata* L. contains functional bioactive compounds such as procyanidins, anthocyanins, a variety of vitamins, and trace elements. These compounds, including proanthocyanidins and anthocyanidins, are widely studied natural plant extracts that possess high antioxidant activity and are non-toxic, making them promising alternatives to antibiotics in production of livestock and poultry. Furthermore, the antioxidative properties of procyanidins and anthocyanins have potential therapeutic applications for oxidative injury in animals and are extensively utilized in the human health products industry, primarily through absorption and action in the intestinal tracts.

Our initial experiment revealed that anthocyanins extracted from *Dioscorea alata* L. exhibit potential for relieving LPS-induced colitis in mice [4]. This experiment establishes a theoretical foundation for the development and application of *Dioscorea alata* L. In the present study, *Dioscorea alata* L. obtained from Hainan province was utilized as the primary raw material, and the crude extracts were obtained through extraction and purification. We established a model of hydrogen peroxide (H_2_O_2_) damage in pig intestinal epithelial cells (IPEC-J2) to investigate the impact of the crude extracts on antioxidant enzyme activity, the expression of membrane transport-related genes, and the phosphorylation level of the NF-κB pathway in IPEC-J2 cells subjected to oxidative stress. Our study aims to explore the protective effects of crude extracts against oxidative stress in IPEC-J2 cells and to provide a theoretical basis for the development and application of *Dioscorea alata* L.

## 2. Materials and Methods

### 2.1. Materials

The present study employed crude extracts derived from *Dioscorea alata L.* cultivated at Hainan University in Danzhou City, Hainan Province. Analysis of the extract revealed that proanthocyanidin B2 was the predominant component with a content of 1852.98 ng/mg, followed by proanthocyanidin B4 with a content of 475.24 ng/mg [5]. The IPEC-J2 cells utilized in this study were kindly provided by Yun Chen, an associate professor at the College of Animal Science and Technology in Hainan University. A 30% solution of H_2_O_2_ was obtained from Xilong Chemical Co., Ltd. (Shantou, China), while fetal bovine serum (FBS) and Dulbecco’s Modified Eagle Medium/Nutrient Mixture with high glucose (DMEM) were purchased from Hyclone (Logan, UT, USA). Cell Counting Kit-8 (CCK-8) Assay Kit was purchased from Biosharp (Hefei, China), and the BCA Protein Assay Kit was purchased from Beyotime Biotechnology Co., Ltd. (Shanghai, China). Total antioxidant capacity (T-AOC, A015-1, λ = 520 nm), total superoxide dismutase (T-SOD, A001-1, λ = 550 nm), catalase (CAT, A007-1-1, λ = 405 nm), glutathione peroxidase (GSH-P_X_, A005, λ = 412 nm), and malondialdehyde (MDA, A003-1, λ = 532 nm) assay kits were obtained from Nanjing Jiancheng Bioengineering Institute (Nanjing, China). The RNA-easy Isolation Reagent Kit (R701), HiScript III RT SuperMix for qPCR(+Gdna wiper) Kit (R323-01), 2×Rapid Tap Master Mix Kit (P222), and ChamQ Universal SYBR qPCR Master Mix Kit (Q711) were procured from Nanjing Vazyme Biotechnology Co., Ltd. (Nanjing, China). The Polyvinylidene difluoride (PVDF) membrane was sourced from Beyotime (Shanghai, China), while the ECL Chemiluminescence substrate (ultra-sensitive) was obtained from Biosharp (Hefei, China).

### 2.2. Cell Culture

The IPEC-J2 cell line was cultivated and sustained in a complete medium, comprising 90% Dulbecco’s modified Eagle’s medium (DMEM) containing high glucose, 10% fetal bovine serum, and 1% penicillin-streptomycin. The cells were incubated at a temperature of 37 °C in a humidified atmosphere enriched with 5% CO_2_.

### 2.3. Assessment of Cell Cytotoxicity

In order to determine the optimal concentration of H_2_O_2_, the IPEC-J2 cells were seeded into 96-well plates as per the aforementioned protocol. Subsequently, different concentrations of H_2_O_2_ (0, 40, 80, 120, 160, and 200 μmol/L) were introduced to the wells and allowed to incubate for 2, 4, 6, 8, and 10 h. Negative controls were treated with PBS in lieu of H_2_O_2_. To remove any residual H_2_O_2_, the cells were washed thrice with PBS. Cell viability was assessed using the Cell Counting Kit-8 (CCK-8) by adding DMEM and CCK-8 solution to each well for a period of 2 h. The resulting data were recorded by measuring the absorbance at 450 nm without delay.

### 2.4. Enzyme Assays

The IPEC-J2 cells were subjected to varying concentrations of H_2_O_2_ based on their survival rates. The experiment aimed to evaluate the pre-protective potential of the crude extracts of *Dioscorea alata* L. against oxidative damage caused by H_2_O_2_ in IPEC-J2 cells. Figure 1 outlines the experimental design, which involved a blank control group and negative control group cultured with DMEM high glucose medium for 8 h. Additionally, a positive control group and Groups T1/T2/T3 were cultured with 50 μg/mL VC and 25, 50, 75 μg/mL crude extracts of *Dioscorea alata* L., respectively, in DMEM high glucose medium for the same duration. These groups were subsequently washed twice with PBS buffer and treated with medium and 120 μmol/L H_2_O_2_, followed by the assessment of antioxidant indexes after 8 h. The cellular activities of T-AOC, T-SOD, CAT, GSH-P_X_, and the MDA content were determined using commercial kits as per the manufacturer’s instructions. The results were analyzed using a Multiskan Spectrum.

### 2.5. Quantitative PCR

The experimental design, as presented in the table above, involved the use of qPCR to evaluate the expression of SOD1, SOD2, GSH-P_X_, NF-κB, SGLT1, and GLUT2 mRNA. The RNA in the cells was extracted utilizing the RNA-easy Isolation Reagent Kit, followed by reverse transcription using cDNA as a template for quantitative analysis as per the instructions of HiScript III RT SuperMix for qPCR (+gDNA wiper) kit. The qPCR reaction was performed using a total volume of 10 μL under the following conditions: 95 °C for 30 s for initial denaturation, followed by 40 cycles of 95 °C for 10 s and 60 °C for 30 s. The relative mRNA expression of the target gene was calculated using the 2^−∆∆Ct^ method. The primer sequences used are provided in Table 1, with their respective gene sequence numbers verified using the NCBI website BLAST. The primers for each gene were designed using Primer 5 software and synthesized by Sangon Biotech Co., Ltd. (Shanghai, China).

### 2.6. Western Blot

Proteins were extracted from IPEC-J2 cells using a lysis buffer (RIPA, Beyotime, Shanghai, China) and quantified with the bicinchoninic acid (BCA, Beyotime) assay. Once protein concentration was measured, they were loaded and separated by 10% SDS-PAGE, and subsequently electrophoretically transferred (Mini-PROTEAN Tetra System; Bio-Rad, Hercules, CA, USA) to a polyvinylidene fluoride (PVDF, Beyotime) membrane. Following this, the membranes were incubated overnight at 4 °C with primary antibodies including IκBα (L35A5) Mouse mAb (Amino-terminal Antigen) (#4814, 39 kDa, 1:1000, Cell Signaling Technology (CST), Danvers, MA, USA), Phospho-IκBα (Ser32) (14D4) Rabbit mAb (#2859, 40 kDa, 1:1000, Cell Signaling Technology (CST)), NF-κB p65 (L8F6) Mouse mAb (#6956, 65 kDa, 1:1000, Cell Signaling Technology (CST)), Phospho-NF-κB p65 (Ser536) (93H1) Rabbit mAb (#3033, 65 kDa, 1:1000, Cell Signaling Technology (CST)), or GAPDH Polyclonal Antibody (10494-1-AP, 36 kDa, 1:5000–1:40,000, Wuhan Sanying Biotechnology Co., Ltd., Wuhan, China). After washing thrice with TBS with Tween-20 (TBST), membranes were incubated with Horseradish Peroxidase-labeled Goat Anti-Mouse IgG (H + L) (A0216, 1:1000, Beyotime, Haimen, China) or Goat Anti-Rabbit IgG (H + L) (A0208, 1:1000, Beyotime) secondary antibodies for 1 h at room temperature. Following another washing with TBST, the immunoreactive bands were visualized using ultra-sensitive ECL chemiluminescence substrate (Biosharp, Tallinn, Estonia). Finally, relative protein levels were quantified using Image-J software.

### 2.7. Statistical Analysis

In the initial phase of the experiment, optimal conditions for inducing oxidative damage in IPEC-J2 cells were determined based on varying survival rates of cells and antioxidant indicators at different time points. Subsequently, the protective effects of crude extracts of *Dioscorea alata* L. were evaluated by comparing results from different treatment groups. Statistical analysis was performed using Excel software, and SPSS 21 software was used for one-way analysis of variance and LSD method was used for multiple comparisons. The data are presented as mean ± standard deviation. *p*-values less than 0.05 were considered statistically significant.

## 3. Results

### 3.1. Effects of H_2_O_2_ Concentration and Treatment Time on Relative Survival Rate of IPEC-J2 Cells

At high concentrations, H_2_O_2_ induces oxidative stress and reduces the survival of IPEC-J2 cells. The effects of H_2_O_2_ on IPEC-J2 cells were illustrated in Figure 2. Viability of IPEC-J2 cells decreased significantly (*p* < 0.01) after treatment with 120 μmol/L H_2_O_2_ for 2, 4, 6, 8 or 10 h. When the duration of H_2_O_2_ treatment was 2 or 4 h and the concentration of H_2_O_2_ was greater than 80 μmol/L, the survival rate of cells significantly decreased compared to the control group (*p* < 0.01). Moreover, the viability of IPEC-J2 cells decreased significantly (*p* < 0.01) after treatment with more than 120 μmol/L H_2_O_2_ for 6 h. When the cells were exposed to H_2_O_2_ for 8 h and the concentration of H_2_O_2_ was greater than 80 μmol/L, the survival rate of IPEC-J2 cells was significantly lower than the control group (*p* < 0.01). Additionally, when the IPEC-J2 cells were incubated with H_2_O_2_ for 10 h and the incubation concentration was enhanced, the survival rate of IPEC-J2 cells was significantly lower. With the passage of time, the survival rate of cells showed a trend of first increase and then decrease after adding 40–160 μmol/L H_2_O_2_, and reached the highest when the cells were injured for 6 h, and the survival rate of cells decreased sharply when the cells were injured for 8 h.

### 3.2. Effects of H_2_O_2_ Concentration and Time on Antioxidative Index of IPEC-J2 Cells

Figure 3 illustrates the temporal profile of total antioxidant capacity (T-AOC) and the activities of catalase (CAT), total superoxide dismutase (T-SOD), and glutathione peroxidase (GSH-P_X_) in IPEC-J2 cells treated with 120 μmol/L H_2_O_2_. It was observed that the activities of T-SOD, and GSH-P_X_ in IPEC-J2 cells were significantly increased after exposure to 120 μmol/L H_2_O_2_ for 8 h compared to the control group (*p* < 0.05). The T-AOC in IPEC-J2 cells exposed to low concentrations (80 or 120 µmol/L) of H_2_O_2_ for 8 h was found to be the highest among the groups treated for 4, 6, and 10 h. However, the T-AOC in IPEC-J2 cells exposed to H_2_O_2_ for 8 h was reduced in a dose-dependent manner, particularly with H_2_O_2_ at a dosage of 160 μmol/L (*p* < 0.01) when compared with the control group. The MDA content did not show a significant difference among the 80, 120, or 160 μmol/L groups when IPEC-J2 cells were exposed to H_2_O_2_ for 8 h (*p* > 0.05). However, the MDA content was significantly higher than that of the control group when IPEC-J2 cells were exposed to 120 μmol/L H_2_O_2_ for 10 h (*p* < 0.05). These findings suggest that H_2_O_2_ exposure induced oxidative stress in IPEC-J2 cells and modulated the antioxidant response in a time- and dose-dependent manner.

### 3.3. Changes in Antioxidant Function of IPEC-J2 Cells under Crude Extract Preprotection

The present study used 120 μmol/L H_2_O_2_ and 8 h of damage time to establish a model of oxidative stress, based on the above findings. The blank control group was treated with high glucose DMEM with double-antibody alone, while the negative control group was supplemented with 120 μmol/L H_2_O_2_ in high glucose DMEM with double-antibody. The positive control group was treated with 50 μg/mL VC for 8 h in high glucose DMEM with double-antibody, then cleaned twice with PBS buffer solution, and 120 μmol/L H_2_O_2_ was added for 8 h. In Group T1, T2, and T3, 50 μg/mL VC was replaced by 25 μg/mL, 50 μg/mL, and 75 μg/mL crude extract, respectively.

The results depicted in Figure 4A–D revealed that the T-AOC, the activity of CAT, T-SOD, and GSH-P_X_ in the negative control group were lower than those in the blank control group. Particularly, the activity of T-SOD in the negative control group was significantly lower than that in the blank control group(*p* < 0.01). Moreover, the T-AOC, the activity of CAT, T-SOD, and GSH-P_X_ in the positive control group were higher than those in the negative control group, and the activity of T-SOD in the positive control group was significantly higher than that in the negative control group.

In addition, Figure 4A–C showed that the T-AOC content, the activity of CAT, and T-SOD in cells increased with the increase in crude extract concentrations. Furthermore, the T-AOC, the activity of CAT, and T-SOD in Group T3 were the highest and significantly different from those in the negative control group and positive control group (*p* < 0.01). Moreover, the T-AOC and the CAT activity in the T3 group were significantly different from the blank control group (*p* < 0.01). There were no significant differences among all groups as shown in Figure 4D. Nevertheless, the MDA content decreased with the increase in crude extract concentration. The MDA content in the negative control group, positive control group, Group T1, T2, and T3 was significantly increased compared with the blank control group (*p* < 0.01). Moreover, the MDA content in Group T1 was significantly higher than that in the positive control group (*p* < 0.01).

### 3.4. Effects of Crude Extracts on Gene Expression of IPEC-J2 Cells Pre-Protected by H_2_O_2_ Injury

The results presented in Figure 5 indicate that the mRNA levels of SOD1 and SGLT1 increased as the concentration of crude extract increased. In contrast, the mRNA levels of GSH-P_X_, NF-κB, and GULT2 decreased as the concentration of crude extract increased. Moreover, the mRNA levels of SOD2 increased initially and then decreased with the increase in crude extract concentration. Notably, the mRNA levels of SOD1 and SGLT1 in Group T3 were significantly higher than those in the negative and positive control groups. The mRNA levels of SOD2 in Group T2 were significantly higher than those in the blank and negative control groups. The mRNA levels of NF-κB in Group T3 were significantly lower than those in the negative control group. The mRNA levels of GULT2 in Group T2 and T3 were significantly lower than those in the blank and negative control groups (*p* < 0.01).

### 3.5. Effects of Crude Extracts on Expression Level of Oxidative Damage Protein in IPEC-J2 Cells Induced by H_2_O_2_

Figure 6 presents the results of the phosphorylation levels of IκB and p65 proteins under different concentrations of crude extract. The phosphorylation level of IκB protein showed an increase with the increase in crude extract concentration. Moreover, the phosphorylation level of IκB in Group T3 was significantly higher than that in the blank control group, negative control group, Group T1 and Group T2 (*p* < 0.01). On the other hand, the phosphorylation level of p65 protein in negative control group, positive control group, Group T1, T2, and T3 was significantly lower than that in blank control group (*p* < 0.01). Additionally, the phosphorylation levels of p65 protein in Group T1, T2 and T3 were significantly higher than those in negative control group, and significantly lower than those in the positive control group (*p* < 0.01).

## 4. Discussion

The cellular response to oxidative stress in animals is often accompanied by apoptosis, as H_2_O_2_ is able to penetrate cytomembranes and induce cellular damage through oxidation. Consequently, the establishment of an H_2_O_2_-induced cellular damage model is crucial for subsequent studies investigating the pre-protective effects of crude extracts of *Dioscorea alata* L. This model is relatively stable and widely used in studies on intestinal inflammation, with important practical significance. For instance, H_2_O_2_ has been utilized to treat IPEC-1 cells [6], IPEC-J2 cells [7,8], Caco-2 cells [9,10], and IDECs cells [11] to construct cellular damage models, where the survival rate of cells reached approximately 50%. The length of the damaged time was determined based on the survival rate of cells, which varied between different cell types. Under certain conditions, the H_2_O_2_-induced cellular damage model can maintain cells in a relatively stable state, where they are most sensitive to external stimuli. In this experiment, IPEC-J2 cells were exposed to varying concentrations of H_2_O_2_ (0, 40, 80, 120, 160, and 200 μmol/L) for different durations (2, 4, 6, 8, and 10 h). The survival rate of the cells was measured and it was found that the survival rate of IPEC-J2 cells was lower than 70% and decreased sharply with increasing duration of exposure. Furthermore, the cells demonstrated no ability for self-recovery. Therefore, it was concluded that the survival rate of IPEC-J2 cells in the most sensitive state was approximately 70%.

After the body produces oxidative damage, cells initiate a stress response that induces undamaged cells or cells with lesser damage to produce antioxidants such as antioxidant enzymes to adjust antioxidant ability. Different cells exhibit varying tolerance to H_2_O_2_, which is mainly reflected in the distinct antioxidant reactions observed when they are damaged, including changes in antioxidant enzyme activity and oxidative information such as MDA. Wijeratne et al. [12] demonstrated that H_2_O_2_ concentration has different effects on various antioxidant enzymes by constructing a human intestinal cell injury model. Specifically, the activity of GSH-P_X_ increased with the rise in H_2_O_2_ concentration, whereas the activity of CAT showed no significant change with the alteration in H_2_O_2_ concentration. H_2_O_2_ can freely diffuse into cells, activate cellular inflammatory signaling pathways, and induce oxidative stress and self-immunization in cells. At low concentrations, intracellular antioxidant factors degrade H_2_O_2_ to H_2_O and O_2_, hampering the establishment of a damage model. Excessive H_2_O_2_ can penetrate intracellular organelles, causing irreversible damage, and is, therefore, unsuitable for constructing a damage model. To determine the optimal test range of cellular viability, the activity of T-AOC, T-SOD, CAT, GSH-P_X_, and MDA content in cells were assessed at different concentrations of H_2_O_2_ (0, 80, 120, and 160 μmol/L) and durations of injury (4, 6, 8, and 10 h). The antioxidant index in each H_2_O_2_-damaged group showed an increasing trend at first, followed by a decreasing trend, with a turning point at 8 h of H_2_O_2_-induced damage. When cells were damaged by 120 μmol/L H_2_O_2_ for 8 h, the activity of each antioxidant enzyme was the highest, and MDA content did not vary with time. The cellular state was most sensitive to H_2_O_2_ damage at 8 h and 120 μmol/L H_2_O_2_, with marked alterations in antioxidant enzyme activity and self-recovery ability. Therefore, these conditions were established as appropriate for the establishment of a damage model, based on the survival rate of cells and antioxidant index.

The measurement of the activity of total antioxidant capacity (T-AOC) serves as an indicator of the overall antioxidant activity of cells. Catalase (CAT), a hydrogen peroxide (H_2_O_2_) scavenger produced in the body, effectively decomposes H_2_O_2_, reducing oxidative stress [13,14]. Superoxide dismutase (SOD), a scavenger produced by the body, protects the structural and functional integrity of cytomembranes and is a critical scavenger of superoxide anion radicals. Glutathione peroxidase (GSH-P_X_), a peroxide-scavenging antioxidant in the body, effectively removes H_2_O_2_ and some organic peroxides, making it an important component of the physical antioxidant defense system [15]. Malondialdehyde (MDA), a lipid peroxidation product in the body, is a crucial indicator for detecting oxidative stress. In this study, three experimental groups (T1, T2 and T3) were set within the range of the pharmacological toxicity concentration of crude extract of *Dioscorea alata* L. The H_2_O_2_-injured model served as the negative control group, while vitamin C (VC) served as the positive control group. The T-AOC activity increased as the concentration of the crude extract of *Dioscorea alata* L. increased, with Groups T2 and T3 showing higher activity levels than the positive and negative control groups. These findings indicate that the total antioxidant capacity of 50 and 75 μg/mL crude extract of *Dioscorea alata L.* on IPEC-J2 cells was superior to VC. CAT activity in Group T1 was lower than that of the control groups, while that in Group T3 was higher than that in the control groups, increasing as the concentration of the crude extract of *Dioscorea alata* L. increased. These results indicate that the crude extract of *Dioscorea alata* L. effectively removes H_2_O_2_, with a better scavenging effect at higher concentrations. At low concentrations of the crude extract, it effectively removed H_2_O_2_, resulting in a concentration of H_2_O_2_ that was lower than that in the negative control group, resulting in lower CAT activity than that of the negative control group. Studies have shown that procyanidins and anthocyanidins can induce the secretion of antioxidant enzymes during scavenging of peroxide free radicals such as H_2_O_2_. Thus, it is probable that the crude extract further stimulates the production of corresponding catalase after scavenging H_2_O_2_, leading to increased CAT activity with increased concentration of crude extract of *Dioscorea alata* L. MDA is the main product of cellular lipid peroxidation, and the content of MDA in experimental groups decreased with an increase in the concentration of the crude extract of *Dioscorea alata* L. These results indicate that the antioxidant capacity of the crude extract increased with increasing concentration; however, the effect was slower than that of VC.

The activity of superoxide dismutase (SOD) was found to be higher in experimental groups treated with crude extract-ing from *Dioscorea alata* L. compared to the positive control group and the negative control group. Glutathione peroxidase (GSH-P_X_) is a major antioxidant enzyme in animal bodies that effectively protects the body from oxidative stress caused by lipid peroxidation. The activity of GSH-P_X_ is directly correlated with the severity of cellular damage. Production and secretion of SOD and GSH-P_X_ are regulated by the expression of corresponding genes. In this study, the mRNA expression levels of the two enzymes were measured and found to be similar to the change in enzyme activity, indicating a negative correlation between the activity of GSH-P_X_ and the degree of cellular damage. In another study, Suong et al. [16] found that replacing Napier grass silage with anthocyanin-rich black cane silage increased the concentrations of GSH-P_X_ in blood of crossbred Thai-native Anglo-Nubian male goats. Procyanidins in the small intestine were absorbed and transported to various parts of the body through blood resulting in increased GSH-P_X_ content. Proanthocyanidins and anthocyanidins are two major bioactive substances in grapes and red wine. Proanthocyanidins can be converted into anthocyanidins in plants through regulation of the white anthocyanidin reductase gene [17]. A study by Jun-Lan Zhang [18] found that the expression level of GSH-P_X_ gene in jejunum was lower in weaned piglets fed with grape seed procyanidins (GSPs) compared to the blank control group. Although IPEC-J2 cells are commonly used as a simple model for in vitro studies, the results may differ from animal tests. In this study, the activity of GSH-P_X_ decreased with increasing crude extract concentration of *Dioscorea alata* L., indicating that the crude extract can effectively remove H_2_O_2_ and produce similar effects as GSH-P_X_. Furthermore, it reduces the degree of cell damage, resulting in a decrease in the expression level of GSH-P_X_ gene and the activity of GSH-P_X_.

The small intestine is the primary site for nutrient absorption in animals, and it can absorb a variety of nutrients present in food. Proanthocyanidins and anthocyanidins contain multiple polymerized structures, including various glycosidic structures, which can be absorbed by the small intestine, leading to their distribution throughout the body. Recent studies have found that the crude extracts of *Dioscorea alata* L. can affect the activity of antioxidant enzymes and the genic expression of IPEC-J2 cells, indicating that the procyanidins and anthocyanins present in the crude extracts can act both extracellularly and intracellularly [19,20,21]. The transport of anthocyanins within and outside the cells involves both passive diffusion and active transport. Passive diffusion occurs through differential concentration inside and outside the cells, which is not saturable and is positively related to the concentration. In contrast, active transport is mediated by carriers, which can become saturated, and their transport efficiency is not linearly related to their concentration, such as anthocyanins galactose glucoside. Sodium-dependent glucose transporters (SGLT) are carriers that actively transport glucose into cells against the concentration gradient under the action of sodium and potassium pumps. The SGLT1 carrier protein is primarily expressed in small intestinal cells and plays a vital role in the absorption of the small intestine [22]. Glucose transporter (GLUT) is a diffusible transmembrane glycoprotein that mediates glucose transport into cells. GLUT2 is mostly expressed in the intestinal tract and has unique characteristics, such as a weak affinity for glucose and strong affinity for glucosamine. In this experiment, the expression level of the SGLT1 gene increased, and the expression level of the GLUT2 gene decreased with the increase in crude extract concentration of *Dioscorea alata* L., indicating that SGLT1 or GLUT2 is involved in the transmembrane transport of procyanidins and anthocyanins, and active transport is the primary mode of transport. Studies have found that the efficiency of absorption of anthocyanins is related to the degree of recognition of corresponding transporters, and the absorption of cyanidin-3-glucoside (C3G) in the small intestine depends on the activity of SGLT1 and GLUT2 [23,24]. The absorptive modes of anthocyanins from different *Dioscorea alata* L. in small intestinal epithelial cells (Caco-2) were all passive diffusion [25]. The transport of anthocyanins from blueberries in Caco-2 cells showed monomeric differences, with the highest absorptive efficiency of Malvidin 3-glucoside and the lowest absorptive efficiency of Delphinidin-3-glucoside. It has been discovered that the bioavailability of anthocyanidins containing glucoside groups is higher than that containing galactoside groups, and their utilization is related to the transport mode [26]. As a result, it is evident that some components of proanthocyanins and anthocyanins present in the crude extract of *Dioscorea alata* L. are transported into cells, resulting in their ability to reduce cellular oxidative damage. However, the specific substances involved in this process require further exploration.

Oxidative stress is a well-established cause of inflammation in cells. The nuclear factor kappa B (NF-κB) plays a crucial role in immune regulation, and its activity can be induced by various factors in cells, such as ROS, tumor necrosis factor α (TNF-α), and lipopolysaccharide (LPS), among others. NF-κB is a member of the Rel transcription factor family, and its p65 subunit is essential for studying the NF-κB signaling pathway. Under normal conditions, the NF-κB signaling pathway of cells remains in a resting state, and the p65-p50 heterodimer specifically binds to IκB protein to form a nonfunctional trimer. Upon cell stimulation, the IκB protein is phosphorylated and separated from the trimer, thereby activating the p65-p50 dimer and consequently, the NF-κB signaling pathway. In macrophages, the p65-p50 dimer regulates most up-regulated genes of NF-κB and enters into the nucleus after activation to stimulate the transcription and expression of NF-κB genes [27]. When cells are stimulated by H_2_O_2_ to produce oxidative damage, they can produce lactate dehydrogenase (LDH), which changes the permeability of cytomembrane and karyotheca, accelerating the entry of the p65-p50 dimer into the nucleus and activating the NF-κB pathway. This activation can occur briefly and quickly to prompt the cells to conduct self-regulation. However, long-term activation of this pathway can lead to the disorder of the internal environment in the body, inducing various diseases such as atherosclerosis, ulcerative colitis, and colon cancer [28,29,30,31,32]. The activation of NF-κB can lead to changes in protein pathways and cause the expression of corresponding genes. The expression of NF-κB genes can be stimulated and enhanced by H_2_O_2_, while Vitamin C (VC) and the crude extract of *Dioscorea alata* L. can inhibit the expression of NF-κB genes. The higher the concentration of crude extract, the stronger its inhibitory ability, indicating that the substances in the crude extract of *Dioscorea alata* L. can remove intracellular oxidative factors, thus reducing the cellular oxidative stress response. In summary, oxidative stress-induced inflammation and the role of NF-κB in immune regulation are critical factors that can lead to various diseases. The activation of NF-κB can have both positive and negative effects on cellular function, and its inhibition can be a potential therapeutic target for mitigating oxidative stress-related diseases. The study of the regulatory mechanisms of NF-κB signaling pathway and the identification of natural compounds that can modulate its activity are crucial areas of research in the field of inflammation and disease prevention.

The NF-κB signaling pathway in cells can be regulated through three mechanisms, as described in previous studies [33,34,35,36,37]. The first mechanism involves the activation of inflammatory pathways by exogenous factors such as TNF-α, IL-1, and IL-6. The second mechanism involves the inhibition of IκB protein phosphorylation, which reduces the decomposition of the signal protein trimer into the active dimer. The third mechanism involves the decomposition of p105 protein into p50 protein, which can form homologous dimer and enter the nucleus for regulation. The regulatory effects of different substances on the NF-κB signaling pathway are distinct. In this study, the protein expression levels of the p-IκB/IκB protein phosphorylation pathway and p-p65/p65 protein phosphorylation pathway were detected using Western blot analysis. The results showed that VC activated both phosphorylation pathways simultaneously, while the extract of *Dioscorea alata* L. activated the phosphorylation of IκB protein and inhibited the phosphorylation of p65 protein, with the degree of phosphorylation being concentration-dependent. VC activated IκB protein and promoted the production of heterodimers, which further promoted the phosphorylation of p65 protein, leading to the decomposition of heterodimers and production of p50 without cross-domain activation domains. This reduced the entry of heterodimers into the nucleus and subsequent activation of the signaling pathway. On the other hand, the crude extract of *Dioscorea alata* L. effectively inhibited the phosphorylation of IκB protein and the activation of the NF-κB signal pathway from the upstream pathway at low concentration. At high concentration, it exhibited a similar inhibitory effect to VC. The activation of the NF-κB signaling pathway is closely related to cellular autoimmunity. The crude extract of *Dioscorea alata* L. can promote the expression of proteins in this pathway after pre-protection of IPEC-J2 cells, indicating that the extract can enhance cellular autoimmunity and activate the self-healing ability of cells under oxidative stress. Overall, these findings suggest that the regulatory mechanisms of NF-κB signaling pathway activation can be modulated by different substances, and the crude extract of *Dioscorea alata* L. has the potential to regulate cellular autoimmunity and self-healing ability under oxidative stress. The results of this study have important implications for the development of new therapeutic strategies for diseases associated with NF-κB signaling pathway dysregulation.

## 5. Conclusions

The 50 μg/mL crude extracts of *Dioscorea alata* L. had the best pre-protective effects on IPEC-J2 cells induced by H_2_O_2_. Additionally, these can reduce cells’ oxidative damage by inhibiting the activation of the NF-κB inflammatory signaling pathway. *Dioscorea alata* L. and its crude extracts can be exploited as natural antioxidants.

## Figures and Tables

**Figure 1 animals-13-01401-f001:**
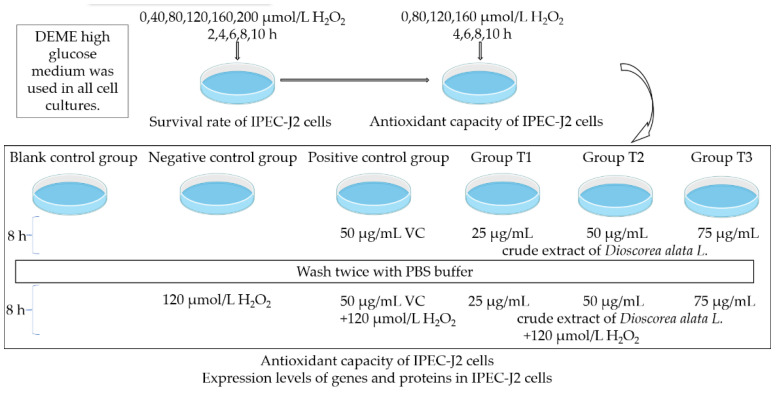
Experimental design.

**Figure 2 animals-13-01401-f002:**
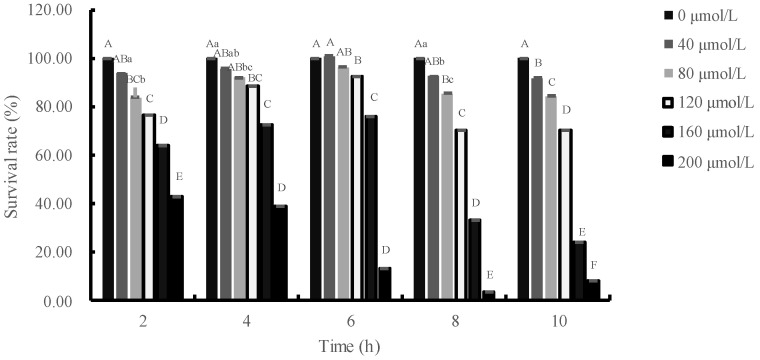
The effects of H_2_O_2_ concentration and action time on the survival rate of IPEC-J2 cells. Note: This figure shows the survival rate of IPEC-J2 cells treated with different concentrations of H_2_O_2_ (0, 40, 80, 120, 160 and 200 μmol/L) at different times (2, 4, 6, 8 and 10 h). At each time point, compared with the 0 μmol/L group, the difference was significant for different lowercase letters (*p* < 0.05), the difference was extremely significant for different uppercase letters (*p* < 0.01), and the difference was not significant for no letters or the same letter shoulder label (*p* > 0.05).

**Figure 3 animals-13-01401-f003:**
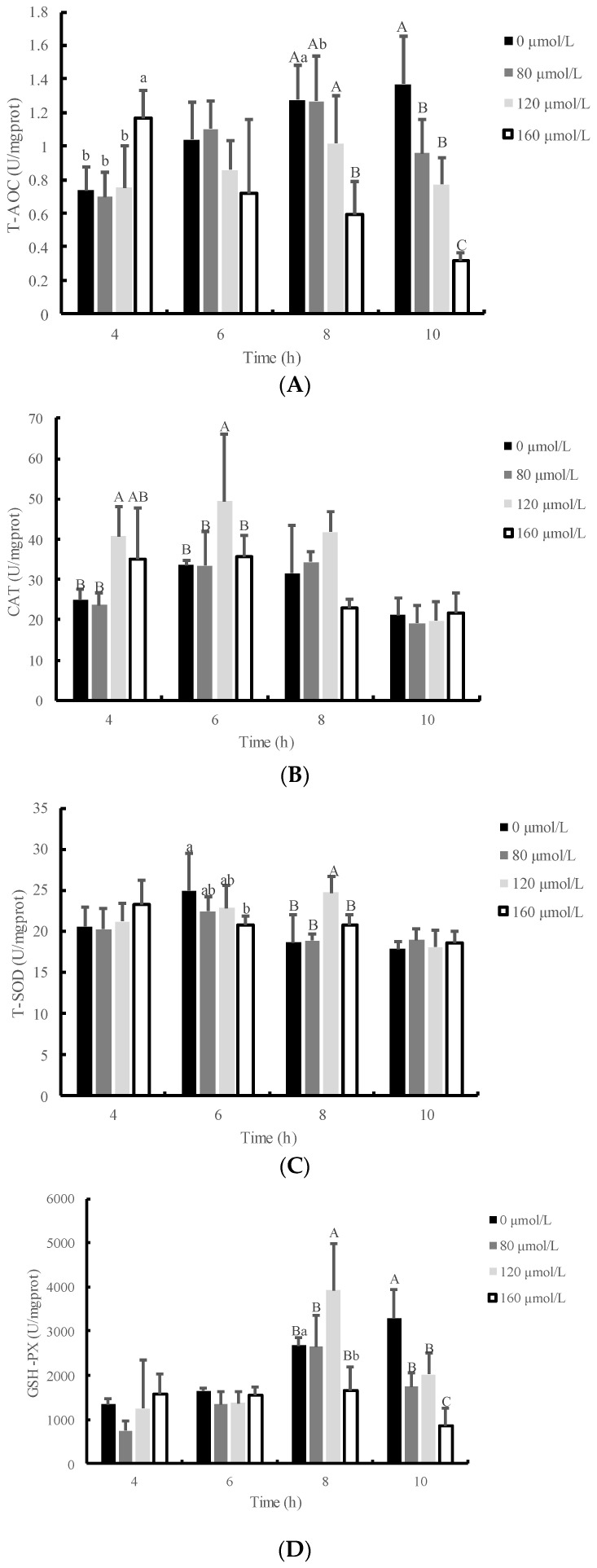
The effects of H_2_O_2_ concentration and action time on the antioxidant capacity of IPEC-J2 cells. Cellular antioxidant capacity is mainly reflected by the following indicators: (**A**) T-AOC, (**B**) CAT activity, (**C**) T-SOD activity, (**D**) GSH-PX activity, (**E**) MDA content. Note: This figure shows the effects of different concentrations of H_2_O_2_ (0, 80, 120 and 160 μmol/L) on the antioxidant performance of IPEC-J2 cells at different times (4, 6, 8 and 10 h). At each time point, compared with the 0 μmol/L group, the difference was significant for different lowercase letters (*p* < 0.05), the difference was extremely significant for different uppercase letters (*p* < 0.01), and the difference was not significant for no letters or the same letter shoulder label (*p* > 0.05).

**Figure 4 animals-13-01401-f004:**
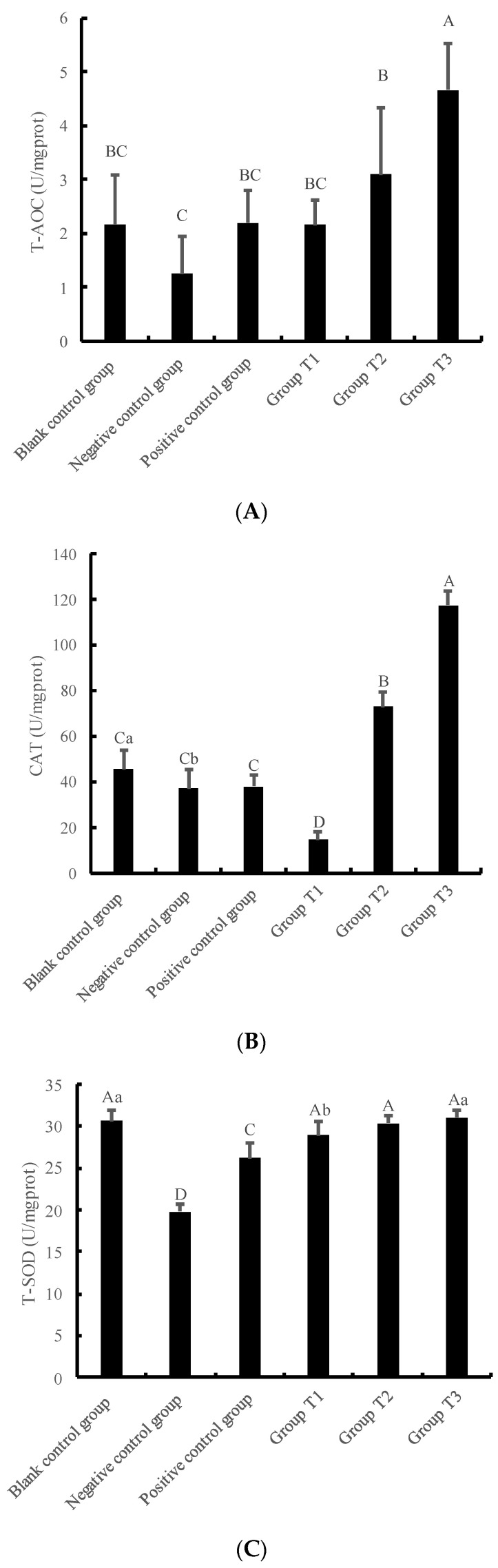
The effects of crude extracts of *Dioscorea alata* L. on antioxidant function of IPEC-J2 cells pre-protected against damage caused by H_2_O_2_. Cellular antioxidant capacity is mainly reflected by the following indicators: (**A**) T-AOC, (**B**) CAT activity, (**C**) T-SOD activity, (**D**) GSH-PX activity, (**E**) MDA content. Note: Compared with the blank control group, the difference was significant for different lowercase letters (*p* < 0.05), the difference was extremely significant for different uppercase letters (*p* < 0.01), and the difference was not significant for no letters or the same letter shoulder label (*p* > 0.05). Blank control group: only DEME high glucose medium; Negative control group: DEME high glucose medium +120 μmol/L H_2_O_2_; Positive control group: DEME high glucose medium +120 μmol/L H_2_O_2_ + 50 μg/mL VC; Group T1-3: DEME high glucose medium +120 μmol/L H_2_O_2_+ crude extracts of *Dioscorea alata* L. at different concentrations (25, 50 and 75 μg/mL).

**Figure 5 animals-13-01401-f005:**
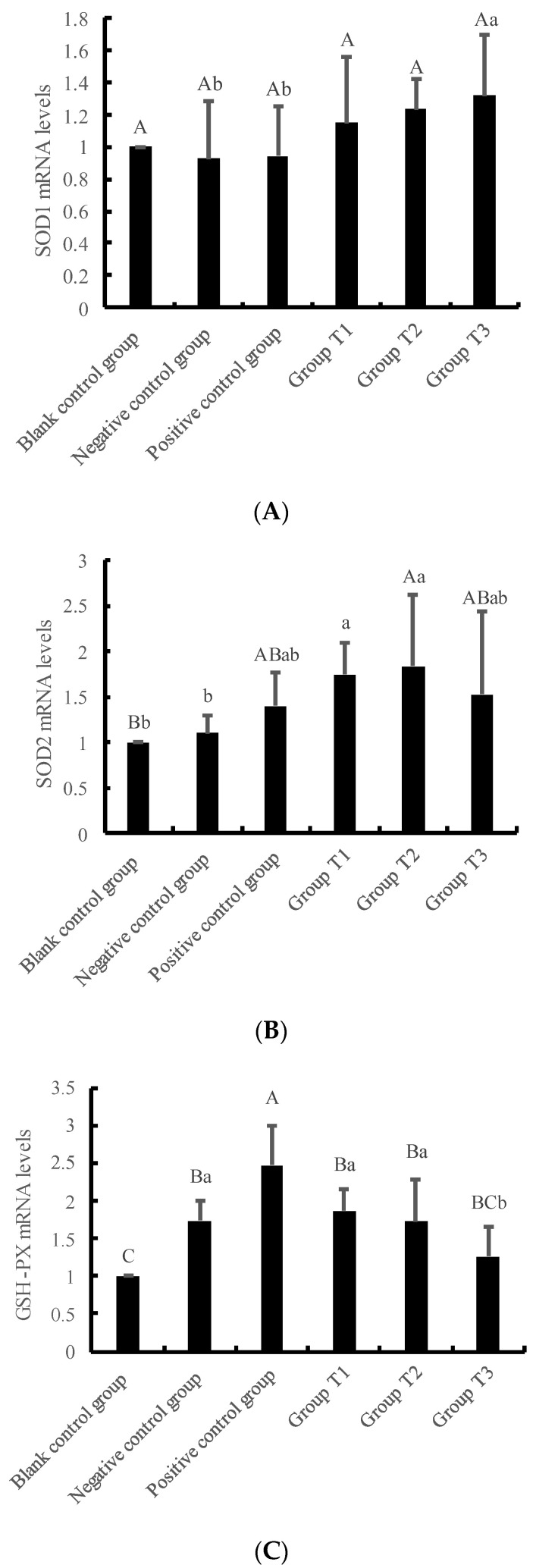
The effects of crude extracts of *Dioscorea alata* L. on gene expression of IPEC-J2 cells pre-protected against injury caused by H_2_O_2_. The preprotective effect of *Dioscorea alata* L. was mainly explored by the expression levels of the following genes: (**A**) SOD1, (**B**) SOD2, (**C**) GSH-PX, (**D**) NF-κB, (**E**) SGLT1, (**F**) GULT2. Note: Compared with the blank control group, the difference was significant for different lowercase letters (*p* < 0.05), the difference was extremely significant for different uppercase letters (*p* < 0.01), and the difference was not significant for no letters or the same letter shoulder label (*p* > 0.05). Blank control group: only DEME high glucose medium; Negative control group: DEME high glucose medium +120 μmol/L H_2_O_2_; Positive control group: DEME high glucose medium +120 μmol/L H_2_O_2_ + 50 μg/mL VC; Group T1–3: DEME high glucose medium +120 μmol/L H_2_O_2_+ crude extracts of *Dioscorea alata L.* at different concentrations (25, 50 and 75 μg/mL).

**Figure 6 animals-13-01401-f006:**
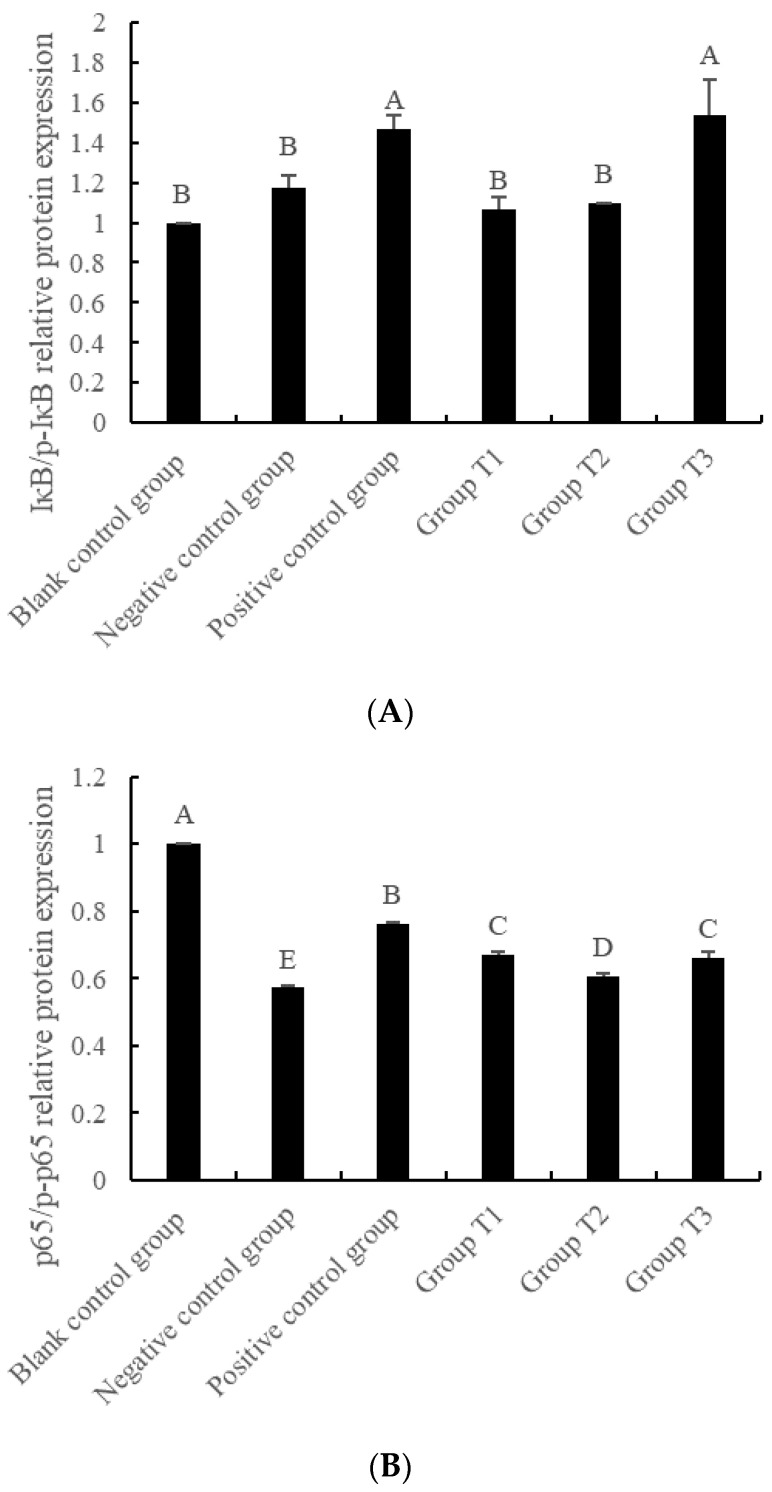
The effects of crude extract of *Dioscorea alata* L. on protein expression level of oxidative damage induced by H_2_O_2_ in IPEC-J2 cells. It is mainly reflected by (**A**) IκB/p-IκB relative protein expression and (**B**) p65/p-p65 relative protein expression. The bands of western blot are shown in (**C**,**D**). Note: Compared with the blank control group, the difference was significant for different lowercase letters (*p* < 0.05), the difference was extremely significant for different uppercase letters (*p* < 0.01), and the difference was not significant for no letters or the same letter shoulder label (*p* > 0.05). Blank control group: only DEME high glucose medium; Negative control group: DEME high glucose medium +120 μmol/L H_2_O_2_; Positive control group: DEME high glucose medium +120 μmol/L H_2_O_2_ + 50 μg/mL VC; Group T1–3: DEME high glucose medium +120 μmol/L H_2_O_2_+ crude extracts of *Dioscorea alata* L. at different concentrations (25, 50 and 75 μg/mL).

**Table 1 animals-13-01401-t001:** Fluorescence quantitative PCR gene primer list.

Gene	Serial Number	Forward Primer Sequence	Reverse Primer Sequence	Size (bp)
GAPDH	NM 001206359.1	CAAGGCTGTGGGCAAGGTCATC	TTCTCCAGGCGGCAGGTCAG	111
NF-κB	NM 001005150	TGGTGTCGCTCTTGTTGAAGTGTG	GCTGCTGTATCCGAGTGCTTGG	108
GSH-P_X_1	NM 214201.1	CGCAATGACATCGCATGGAACTTC	CACTGCTAGGCTCCTGGGACAG	139
SOD1	NM 001190422.	GAAGATTCTGTGATCGCCCTCTCG	TTCATTTCCACCTCTGCCCAAGTC	99
SOD2	NM 214127.2	TGTATCCGTCGGCGTCCAAGG	TCCTGGTTAGAACAAGCGGCAATC	93
SGLT1	NM 001164021.1	TCATCATCGTCCTGGTCGTCTCC	TGAATGTCCTCCTCCTCTGCATCC	130
GLUT2	NM 00109741	TGCTCTGGTCTCTGTCTGTGTCC	ATTCTTCCAAGCCGATCTCCAAGC	91

## Data Availability

All raw data in this study are available from the corresponding author upon reasonable request.

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
