# Peer review of "Pre-Protection and Mechanism of Crude Extracts from Dioscorea alata L. on H2O2-Induced IPEC-J2 Cells Oxidative Damage"

_animals, 2023, doi:10.3390/ani13081401_

Round 1

Reviewer 1 Report

This study aimed to evaluate, in vitro, the damage effect of different doses of H2O2 on intestinal porcine epithelial cells, and under the potential protective action of crude extract (at different doses) of Dioscorea alata L.

The manuscript present scientific soundness, and the experimental design well conceived.  Nevertheless,  3 major issue need be corrected: 1)Crude extract was not defined/characterized in M&M; 2) The statistical analysis (between groups and between time) was not described; 3) An English language edition is essential to improve the readability of the manuscript between L5 and L70 (simple summary, abstract and introduction section). The remaining manuscript is well written.

As minor issues, the Dioscorea alata need be italicized.  A legend for figures should be inserted (p value for different letters and the bars as SD).

Author Response

A response letter to reviewer

Dear reviewer:

Thank you very much for the opportunity to revise the manuscript, we appreciate the reviewer very much for their positive and constructive comments on our manuscript entitled " Pre-protection and mechanism of crude extracts from Dioscorea alata L. on H2O2-induced IPEC-J2 cells oxidative damage ". We have tried our best to revise our manuscript according to the comments and would like to re-submit it for your consideration. Details please refer to the following table. All revisions to the manuscript were marked up using the “Track Changes” function.

We have carefully revised the manuscript according to the comments. We would like to express our greatest appreciation to you and the reviewer for comments on our paper. We hope the manuscript is found to be satisfactory for publication in your journal. If you have any questions about this paper, please do not hesitate to contact us. We hope that the revised version of the manuscript is now acceptable for publication in your journal.

Sincerely,

Yanhong Yun

Huiyu Shi, PhD

Reviewer’s comments:

Describing all changes

1)Crude extract was not defined/characterized in M&M;

Thank you very much for your valuable comments, and the description of the crude extract has been added to the M&M.(Line 80-84)

2) The statistical analysis (between groups and between time) was not described;

We have added descriptive analyses of the trial groups and over time. In the early stage of the experiment, the optimal conditions of IPEC-J2 cells oxida-tive damage model were explored by different levels of survival rate of cells and anti-oxidant indicators at different times. Later, the pre-protective effects of crude extract of Dioscorea alata L. were explored by comparing the results of different treatment groups.(Line 115-123)

3) An English language edition is essential to improve the readability of the manuscript between L5 and L70 (simple summary, abstract and introduction section).

We have perfected the three parts of summary, abstract and introduction to improve the level of English expression (Line 8-76)

4) As minor issues, the Dioscorea alata need be italicized. A legend for figures should be inserted (p value for different letters and the bars as SD).

We have set all “Dioscorea alata L. in this paper to italic format. And we modified the presentation of P values and detailed what they represent under each figure.

Reviewer 2 Report

The manuscript titled “Pre-protection and mechanism of crude extracts from Dioscorea alata L. on H2O2-induced IPEC-J2 cells oxidative damage” investigated the potential protective effect of the crude extracts from Dioscorea alata L., and revealed the possible mechanism. The topic investigated is of significant interest for animal nutrition. A good amount of analytical evaluations have been done. The description of methodology results accurate and quite precise. Therefore, I think it can be accepted. However, there are some question also should be addressed.

1. The author should carefully review the manuscript again to standardize the writing format, for example Line 94, CO2; Line 164, H2O2……

2. The conclusion should be reduced. A detailed "Conclusion" should be provided to state the final result that the authors have reached. Please note you only need to place your conclusion and not keep putting results, because these have already been presented in the manuscript.

3. Author(s) should re-format the references based on journal format. See the instructions for authors.

4. In figure 1 the title of the Y-axis should not be %

5. If the authors tested the intracellular activity of the antioxidant enzymes, the unite should be u/mg. Please check it.

6. The title of the Y-axis in Figure 2 (D) is not complete, same with Figure 3.

7. Figure or Fig should keep consistent in the whole manuscript.

Author Response

A response letter to reviewer

Dear reviewer:

Thank you very much for the opportunity to revise the manuscript, we appreciate the reviewer very much for their positive and constructive comments on our manuscript entitled " Pre-protection and mechanism of crude extracts from Dioscorea alata L. on H2O2-induced IPEC-J2 cells oxidative damage ". We have tried our best to revise our manuscript according to the comments and would like to re-submit it for your consideration. Details please refer to the following table. All revisions to the manuscript were marked up using the “Track Changes” function.

We have carefully revised the manuscript according to the comments. We would like to express our greatest appreciation to you and the reviewer for comments on our paper. We hope the manuscript is found to be satisfactory for publication in your journal. If you have any questions about this paper, please do not hesitate to contact us. We hope that the revised version of the manuscript is now acceptable for publication in your journal.

Sincerely,

Yanhong Yun

Huiyu Shi, PhD

Reviewer’s comments:

Describing all changes

1. The author should carefully review the manuscript again to standardize the writing format, for example Line 94, CO2; Line 164, H2O2……

Thank you very much for your affirmation of this manuscript and your valuable comments. In addition, we have checked the words in the full text and standardized the writing of CO2, H2O2, etc.

2. The conclusion should be reduced. A detailed "Conclusion" should be provided to state the final result that the authors have reached. Please note you only need to place your conclusion and not keep putting results, because these have already been presented in the manuscript.

We have simplified the conclusions to highlight the findings of this study.(Line 548-551)

3. Author(s) should re-format the references based on journal format. See the instructions for authors.

We have modified the format of the references according to the format requirements of the journals.(Line568-652)

4. In figure 1 the title of the Y-axis should not be %

We have carefully checked and modified the Y-axis in the figure 1.(Line 185)

5. If the authors tested the intracellular activity of the antioxidant enzymes, the unite should be u/mg. Please check it.

It is stated in the kit that U/mL and nmol/mL are used to represent the antioxidant indexes in serum. We have checked the whole text and changed the units of T-AOC, CAT, T-SOD and GSH-PX to U/mgprot, and the unit of MDA to nmol/mgprot according to the kit instruction.

6. The title of the Y-axis in Figure 2 (D) is not complete, same with Figure 3.

We have supplemented the Y-axis headings in Figures 2 (D) and 3 (D).(Line 215, Line 259)

7. Figure or Fig should keep consistent in the whole manuscript.

We've unified the way we write "Figure".

Reviewer 3 Report

The manuscript reported the potential preventative effects of crude extracts from Dioscorea alata L. on H2O2-induced oxidative damage using the IPEC-J2 cells. The results are promising, however, major revisions are required before acceptance.

My comments are listed below:

(1) The major composition and levels of active components in crude extracts used should be determined since different plant sources generally have great differences in the active components.

(2) The figure legends should include more details to enhance self-explanatory. For example, the specific treatment of the negative control or positive control shown in figure 3 can not be easily understood by the reader.

(3) A brief figure to indicate the major experiment design is suggested.

(4) Figures of western blot and the results in figure 5 may be wrongly explained by the histogram, please carefully double-check. In another word, figure 5A currently shows the results in Figure 5D; while Figure 4B shows the results in Figure 5C. In addition, if the protein level in control groups has been standardized into level 1, then the data should be relative protein levels, instead of the protein level marked in the current version.

(5) Figure quality and English writing in the manuscript need great improvement.

Author Response

A response letter to reviewer

Dear reviewer:

Thank you very much for the opportunity to revise the manuscript, we appreciate the reviewer very much for their positive and constructive comments on our manuscript entitled " Pre-protection and mechanism of crude extracts from Dioscorea alata L. on H2O2-induced IPEC-J2 cells oxidative damage ". We have tried our best to revise our manuscript according to the comments and would like to re-submit it for your consideration. Details please refer to the following table. All revisions to the manuscript were marked up using the “Track Changes” function.

We have carefully revised the manuscript according to the comments. We would like to express our greatest appreciation to you and the reviewer for comments on our paper. We hope the manuscript is found to be satisfactory for publication in your journal. If you have any questions about this paper, please do not hesitate to contact us. We hope that the revised version of the manuscript is now acceptable for publication in your journal.

Sincerely,

Yanhong Yun

Huiyu Shi, PhD

Reviewer’s comments:

Describing all changes

(1) The major composition and levels of active components in crude extracts used should be determined since different plant sources generally have great differences in the active components.

Thank you very much for your valuable comments. We have explained the main components and contents of the crude extract of Dioscorea alata L. in the article. The detection showed that the most important component in the crude extract of Dioscorea alata L. was proanthocyanidin B2, and its content was 1852.98 ng/mg, followed by proanthocyanidin B4, and its content was 475.24 ng/mg (Wang, 2021). (Line 80-84)

(2) The figure legends should include more details to enhance self-explanatory. For example, the specific treatment of the negative control or positive control shown in figure 3 can not be easily understood by the reader.

Thank you for your question. For this reason, we have checked and revised the full text, and have supplemented the annotations of all figures, including the groups, as detailed in the paper.

(3) A brief figure to indicate the major experiment design is suggested.

Relevant design methods describing the trial have been added.(Line 115-127)

(4) Figures of western blot and the results in figure 5 may be wrongly explained by the histogram, please carefully double-check. In another word, figure 5A currently shows the results in Figure 5D; while Figure 4B shows the results in Figure 5C. In addition, if the protein level in control groups has been standardized into level 1, then the data should be relative protein levels, instead of the protein level marked in the current version.

Thank you for your comments, we have adjusted the order of the pictures, and we did divide the expression level of the target protein by the expression level of the housekeeping protein, so that the protein expression levels of different groups could be compared. However, such data processing methods are used in general papers, and they are all described as protein expression levels, and few are expressed by relative protein levels. (Line 327,331)

(5) Figure quality and English writing in the manuscript need great improvement.

We have tried our best to optimize the related graphs and we have checked our manuscript  by a native English-speaking colleague for improving the level of English expression.

Wang Y.Y. Study on the protective effect and mechanism of crude extract of Dioscorea alata L. on oxidative damage induced by H2O2 in IPEC-J2 cells. Hainan University 2021.

Round 2

Reviewer 1 Report

Dear authors,

Thanks for providing this revised version. The comments and suggestions were adressed. Just a point need be clarified: the significance (p-values) of differences between groups/time was provided by what test?

Author Response

A response letter to reviewer

Dear reviewer:

Thank you very much for the opportunity to revise the manuscript, we appreciate the reviewer very much for their positive and constructive comments on our manuscript entitled " Pre-protection and mechanism of crude extracts from Dioscorea alata L. on H2O2-induced IPEC-J2 cells oxidative damage ". We have tried our best to revise our manuscript according to the comments and would like to re-submit it for your consideration. Details please refer to the following table. All revisions to the manuscript were marked up using the “Track Changes” function.

We have carefully revised the manuscript according to the comments. We would like to express our greatest appreciation to you and the reviewer for comments on our manuscript. We hope the manuscript is found to be satisfactory for publication in your journal. If you have any questions about this manuscript, please do not hesitate to contact us. We hope that the revised version of the manuscript is now acceptable for publication in your journal.

Sincerely,

Yanhong Yun

Huiyu Shi, PhD

Reviewers' comments:

Describing all changes

Reviewer#1: General comments:

Just a point need be clarified: the significance (p-values) of differences between groups/time was provided by what test?

Thank you very much for your comments. SPSS 21 software was used for single factor analysis, and LSD method was used for multiple comparisons. P<0.05 indicated that the difference was significant, and P<0.01 indicated that the difference was extremely significant.(Line 170-172)

Reviewer 3 Report

The authors did not sufficiently address the comments raised at 1st round.

Author Response

A response letter to reviewer

Dear reviewer:

Thank you very much for the opportunity to revise the manuscript, we appreciate the reviewer very much for their positive and constructive comments on our manuscript entitled " Pre-protection and mechanism of crude extracts from Dioscorea alata L. on H2O2-induced IPEC-J2 cells oxidative damage ". We have tried our best to revise our manuscript according to the comments and would like to re-submit it for your consideration. Details please refer to the following table. All revisions to the manuscript were marked up using the “Track Changes” function.

We have carefully revised the manuscript according to the comments. We would like to express our greatest appreciation to you and the reviewer for comments on our manuscript. We hope the manuscript is found to be satisfactory for publication in your journal. If you have any questions about this manuscript, please do not hesitate to contact us. We hope that the revised version of the manuscript is now acceptable for publication in your journal.

Sincerely,

Yanhong Yun

Huiyu Shi, PhD

Reviewers' comments:

Describing all changes

Reviewer#3: General comments:

The authors did not sufficiently address the comments raised at 1st round.

Thank you very much for your comments. We have made further and careful modifications according to the first comments.

(1) The major composition and levels of active components in crude extracts used should be determined since different plant sources generally have great differences in the active components.

Thank you very much for your valuable comments. We have explained the main components and contents of the crude extract of Dioscorea alata L. in the article. The detection showed that the most important component in the crude extract of Dioscorea alata L. was proanthocyanidin B2, and its content was 1852.98 ng/mg, followed by proanthocyanidin B4, and its content was 475.24 ng/mg (Wang, 2021). (Line 82-85)

(2) The figure legends should include more details to enhance self-explanatory. For example, the specific treatment of the negative control or positive control shown in figure 3 can not be easily understood by the reader.

Thank you for your valuable advice. In response to this problem, the full manuscript was checked and revised again, and the annotations of all figures, including grouping, were supplemented.

(3) A brief figure to indicate the major experiment design is suggested.

We think the suggestion you put forward is very valuable, so we have adopted your suggestion and changed the original three-line table into a more intuitive diagram. (Line 128-129)

(4) Figures of western blot and the results in figure 5 may be wrongly explained by the histogram, please carefully double-check. In another word, figure 5A currently shows the results in Figure 5D; while Figure 4B shows the results in Figure 5C. In addition, if the protein level in control groups has been standardized into level 1, then the data should be relative protein levels, instead of the protein level marked in the current version.

Thank you for your comments. We have adjusted the order of the pictures and corrected the expression with "relative protein expression level". (Line 347,350)

(5) Figure quality and English writing in the manuscript need great improvement.

We have tried our best to optimize the related graphs and we have checked our manuscript  by a native English-speaking colleague for improving the level of English expression again.

Wang Y.Y. Study on the protective effect and mechanism of crude extract of Dioscorea alata L. on oxidative damage induced by H2O2 in IPEC-J2 cells. Hainan University 2021.